# Chemokine-Cytokine Networks in the Head and Neck Tumor Microenvironment

**DOI:** 10.3390/ijms22094584

**Published:** 2021-04-27

**Authors:** Sabah Nisar, Parvaiz Yousuf, Tariq Masoodi, Nissar A. Wani, Sheema Hashem, Mayank Singh, Geetanjali Sageena, Deepika Mishra, Rakesh Kumar, Mohammad Haris, Ajaz A. Bhat, Muzafar A. Macha

**Affiliations:** 1Molecular and Metabolic Imaging Laboratory, Cancer Research Department, Sidra Medicine, Doha 26999, Qatar; snisar1@sidra.org (S.N.); shashem@sidra.org (S.H.); mharis@sidra.org (M.H.); 2Department of Zoology, School of Life Sciences, Central University of Kashmir, Ganderbal 191201, India; saleemparvaiz444@gmail.com; 3Department of Genomic Medicine, Genetikode 400102, India; tahamasoodi@gmail.com; 4Department of Biotechnology, School of Life Sciences, Central University of Kashmir, Ganderbal 191201, India; waninh@yahoo.co.in; 5Departmental of Medical Oncology, Dr. B. R. Ambedkar Institute Rotary Cancer Hospital, All India Institute of Medical Sciences, New Delhi 110029, India; mayank.osu@gmail.com; 6Keshav Mahavidyalaya, University of Delhi, New Delhi 110034, India; geetanjalisageena@gmail.com; 7Centre for Dental Education and Research, Department of Oral Pathology and Microbiology, All India Institute of Medical Sciences, New Delhi 110029, India; deepika1904@gmail.com; 8Centre for Advanced Research, School of Biotechnology and Indian Council of Medical Research, Shri Mata Vaishno Devi University, Katra 182320, India; Kumar.rakesh@smvdu.ac.in; 9Laboratory Animal Research Center, Qatar University, Doha 2713, Qatar; 10Watson-Crick Centre for Molecular Medicine, Islamic University of Science and Technology, Awantipora 192122, India

**Keywords:** head and neck squamous cell carcinomas, cytokines, chemokines, tumor microenvironment, apoptosis, invasion, metastasis, angiogenesis, response to therapy, immune evasion

## Abstract

Head and neck squamous cell carcinomas (HNSCCs) are aggressive diseases with a dismal patient prognosis. Despite significant advances in treatment modalities, the five-year survival rate in patients with HNSCC has improved marginally and therefore warrants a comprehensive understanding of the HNSCC biology. Alterations in the cellular and non-cellular components of the HNSCC tumor micro-environment (TME) play a critical role in regulating many hallmarks of cancer development including evasion of apoptosis, activation of invasion, metastasis, angiogenesis, response to therapy, immune escape mechanisms, deregulation of energetics, and therefore the development of an overall aggressive HNSCC phenotype. Cytokines and chemokines are small secretory proteins produced by neoplastic or stromal cells, controlling complex and dynamic cell–cell interactions in the TME to regulate many cancer hallmarks. This review summarizes the current understanding of the complex cytokine/chemokine networks in the HNSCC TME, their role in activating diverse signaling pathways and promoting tumor progression, metastasis, and therapeutic resistance development.

## 1. Introduction

Head and neck squamous cell carcinoma (HNSCC) is a very aggressive disease with a dismal prognosis. With an annual incidence of ~800,000 new cases and 350,000 deaths worldwide, HNSCC is the sixth most common cancer globally [1]. HNSCC includes tumors of the oral cavity, hypopharynx, oropharynx, larynx and, paranasal sinuses and is clinically, pathologically, phenotypically, and biologically a heterogeneous disease [2]. Oral squamous cell carcinoma (OSCC), being the primary subtype of HNSCC, accounts for two-thirds of the cases occurring in developing nations. Although tobacco and alcohol consumption account for nearly 75% of the total HNSCC cases, there has been a recent rise in the incidence of Human Papilloma Virus (HPV) associated oropharynx cancers (OPC) [3]. Cytokines and chemokines are soluble, low molecular weight secretory proteins, which regulate lymphoid tissue development, immune and inflammatory responses by controlling immune cell growth, differentiation, and activation [4,5]. While the cytokines are non-structural, pleiotropic proteins or glycoproteins, which have a complex regulatory influence on inflammation and immunity, chemokines are a large family of low molecular weight (8–14 KDa) heparin-binding chemotactic cytokines that regulate leukocyte trafficking, development, angiogenesis, and hematopoiesis [4,5]. Based on the variations in the structural motif of the first two closely paired and highly conserved cysteine residues, chemokines are divided into CXC, CC, CX3C, and the C subfamilies. While the C subfamily has only two cysteine residues, CXC, CC, and CX3C have four cysteine residues [6]. The letter ‘‘L” followed by a number denotes a specific chemokine (e.g., CCL2 or CXCL8). The receptors are labeled by the letter R followed by the number (e.g., CCR2 or CXCR1) [7,8]. Based on the conserved glutamic acid-leucine-arginine “Glu-Leu-Arg” (ELR) motif at the NH2 terminus, the CXC chemokine family is further subdivided into ELR^+ve^ and ELR^−ve^. The ELR^+ve^ CXC chemokines are angiogenic and activate CXCR2 mediated signaling pathway in endothelial cells, while the ELR^−ve^ CXC chemokines are angio-static and are potent chemo-attractants for mononuclear leukocytes [9,10,11]. Cytokines such as TNF (α and β), interleukin 1 family (IL-1α, IL-1β, IL-1 receptor antagonistic (IL-Ira) and IL-2, IL -6, IL-8, IL-10, IL-11, IL-12, IL-15, IL-16, IL -17, IL-18, IL-19, IL-20, IL-21, IL-22, IL-23, IL-24, interferon’s (α, β, and γ), TGFβ, are produced by various types of cell including mononuclear phagocytic cells, T-lymphocytes, B-lymphocytes, Langerhans cells, polymorph nuclear neutrophils (PMNs), and mast cells [12]. Based on their biological properties, these cytokines are classified into T-helper 1 (Th1), T-helper 2 (Th2), and T-helper 17 (Th17) [13]. While Th1 and Th2 cytokines stimulate cellular and humoral immune responses, Th17 cytokines are known to regulate inflammatory responses and autoimmunity [13]. 

In addition to regulating immune cell function, recent studies have shown that cytokines and chemokines play an important role in cancer-related inflammation and immune evasion processes [14] and help in the development and progression of many tumors including HNSCC [15,16,17]. For example, cytokines/chemokines and growth factors like epidermal growth factor (EGF), IL-1α, IL-1β, IL-6, IL-8, TNF-α, TGF-β, RANTES (CCL5), fibroblast growth factor (FGF), monocyte chemo-attractant protein 1 (MCP-1), tumor necrosis factor (TNF), family granulocyte-macrophage colony-stimulating factor (GM-CSF), vascular endothelial growth factor (VEGF), and hepatocyte growth factor (HGF), are upregulated in the HNSCC tumor micro-environment (TME) and are involved in the progression and metastasis [18,19]. These cytokines and chemokines induce cellular transformation [20], control autocrine or paracrine communication within and between the individual cells in the TME [21], and play diverse roles in the HNSCC by controlling processes not limited to Epithelial–Mesenchymal Transition (EMT), anoikis resistance, invasion and metastasis, angiogenesis and development of therapeutic resistance [22], thus, contributing to the development of aggressive HNSCC tumors. These cytokines and chemokines also create an immunosuppressive TME and help evade anti-tumor immune response [21].

## 2. HNSCC Tumor Microenvironment

The HNSCC TME is a heterogeneous complex of cellular and non-cellular components that dictate aberrant tissue function and promote the development of aggressive tumors [23]. While the non-cellular components include extracellular matrix (ECM) proteins and many physical and chemical parameters, cellular components of HNSCC TME includes immune cells such as T cells, B cells, natural killer cells (NK cells), langerhans cells, dendritic cells (DC), myeloid-derived suppressor cells (MDSCs), macrophages, tumor associated-platelets (TAPs), mast cells, adipocytes, neuroendocrine cells, blood lymphatic vascular cells, endothelial cells (EC), pericytes and cancer-associated fibroblasts (CAFs) [24]. In addition to providing intermediate metabolites and nutrients to the tumor cells, these stromal cells secrete a diverse array of cytokines, chemokines, and growth factors that support tumor growth, progression, metastasis [25], host immunosuppression [14], and promote the development of aggressive tumors [22] (Figure 1). However, dysfunctional T-cells, regulatory T cells (Tregs), MDSCs, impaired NK cell activity, and type 2 macrophages (M2) present in the HNSCC TME have an inverse function and promote tumor growth, metastasis, and resistance to therapy [26]. The immunosuppressive HNSCC TME is also facilitated by the downregulation of MHC molecules (human leukocyte antigen, HLA), inactivation of the antigen processing machinery (APM), and dysregulation of checkpoint proteins (reviewed in [27]). The important HNSCC-associated TME cells, cytokines, chemokines, and growth factors are discussed below.

Cancer-associated Fibroblasts (CAFs) are the major cell type in the HNSCC and help maintain a favorable TME aiding tumorigenesis [28]. Though controversial, CAFs are believed to be generated from myofibroblasts, transformed cancer cells, epithelial cells via epithelial-mesenchymal transition (EMT), resting resident fibroblasts or pericytes via mesothelial-mesenchymal transition (MMT), endothelial cells via endothelial to mesenchymal transition (EndMT), adipocytes, and bone marrow-derived mesenchymal stem cells (MSCs) [29]. HNSCC CAF’s secrete a wide variety of cytokines (autocrine or paracrine in function) and tumor-promoting factors essential for inflammation, cell proliferation, tumor growth, invasion & metastasis, angiogenesis, cancer stem cell (CSC) maintenance, and resistance to therapy [30]. These include various cytokines, interleukins (ILs) such as IL-6, IL-17A, and IL-22, growth factors such as EGF, HGF, VEGF, chemokines such as C-X-C motif chemokine ligands (CXCLs), CXCL1, CXCL8, CXCL12 (SDF-1α), and CXCL14, and C-C motif chemokine ligands (CCLs), CCL2, CCL5 and CCL7 [31,32]. These factors promote ECM degradation and modulation by secreting matrix metalloproteins (MMPs) such as MMP-2 and MMP-9 for effective invasion and metastasis of tumor cells [33]. Endothelins (iso-peptides) produced by vascular epithelium upon binding to CAFs activate ADAM17 and trigger release of EGFR ligands such as amphiregulin and TGF-α [33]. These ligands activate EGFR signaling in HNSCC cells, upregulate COX-2 and stimulate the growth, invasion, and metastasis of HNSCC cells [34]. Although little is known about the interaction of CAF-tumor cells in HNSCC, poor overall survival (OS) of HNSCC patients has been associated with increased α-SMA expression regardless of the clinical stage [30]. All these findings ascertain the credibility of CAFs in promoting growth and thus can be useful in facilitating the development of new therapeutic strategies against tumor progression in HNSCC.

Macrophages engage in both innate and adaptive immune responses and protect the body against invading pathogens. These macrophages can either help tumor growth or destroy tumor cells depending upon the external cues from TME. In response to interferons, macrophages are polarized and activated into pro-inflammatory classical M1 type that produces cytokines, such as interferon-γ (IFN-γ), tumor necrosis factor-α (TNF-α), IL-23, IL-12, CCL5, CXCL9, CXCL10, and CXCL5 and help destroy tumor cells via activating Th1 cells [35]. M2 type macrophages closely resemble tumor-associated macrophages (TAMs) and are characterized by increased expression of IFN-γ, CCL2, CCL5, CXCL16, CXCL10, CXCL9, TNF-α, MMP9 and IL-10, arginase-1, and peroxisome proliferator-activated receptor-γ (PPAR-γ) [36]. HNSCC tumors with high M2 TAM infiltration have an advanced stage lymph node metastasis, and poor patient outcome [37]. Elevated CD68**^+^** macrophages are also associated with poor patient survival [38]. Furthermore, increased M2 TAM infiltration is associated with increased tissue levels of macrophage migration inhibitory factor (MIF) and serum TGF-β levels. Though TGF-β is suppressive in function, MIF helps recruit neutrophils to the TME and promotes invasion and metastasis by producing ROS, MMP9 and, VEGF expression [39]. All these studies conclusively established the pro-tumorigenic role of M2 TAMs in regulating cell proliferation, invasion & metastasis, angiogenesis, and promoting immune evasion.

Neutrophils are the most abundant granulocytes present in the blood and an important component of innate and adaptive immunity by regulating T cell activation, antigen presentation and T cell-independent antibody responses [40]. Like TAMs, tumor-associated neutrophils (TANs) can be either tumor-promoting (N2) or tumor suppressors (N1). By activating platelets, neutrophils enhance the risk of cancer-associated venous thromboembolism (VTE) and death in HNSCC patients [41]. Due to the lack of specific markers, identification and characterization of TANs are difficult. However, nonspecific markers including CD14, CD15, CD16, CD11b, CD62L and CD66b are routinely used for their isolation and characterization (reviewed [37]). Natural Killer (NK) cells are large granular CD3**^−ve^** cytotoxic type 1 innate lymphoid cells that detect and kill virus-infected and cancer cells. Based on the expression of adhesion molecules CD56 and the low-affinity FcγR CD16, NK cells are classified into a highly cytotoxic CD56^low^CD16^high^ population predominantly present in the peripheral blood, and less cytotoxic CD56^high^CD16^low^ cells present in the secondary lymphoid and other tissues [42]. These CD56^high^CD16^low^ NK cells, like neutrophils and macrophages, kill cells directly by secreting a plethora of immunomodulatory molecules such as IL-5, IL-8, IL-10, IL-13, CCL2, CCL3, CCL4, CCL5 IFN-γ, TNF-α, GM-SCF and, CXCL10 [43]. Recently, tumor-infiltrating NK cells from HNSCC patients have been shown to possess a decreased expression of activating receptors like NKG2D, DNAM-1, NKp30, CD16, and 2B4 and upregulation of inhibitory receptors NKG2A and PD-1 compared to NK cells from matched peripheral cells [44]. The study also observed low cytotoxicity and reduced IFN-γ secretion from tumor-infiltrating NK cells in vitro [44]. Though no stimulation is needed for NK activation, a small percentage of NK T-cells (NKT) require priming for activation [45]. These NKT cells are specialized cells with morphological and functional characteristics and surface markers of both T and NK cells. The presence of another small subset of invariant NK T cells (iNKT) that express invariant αβ T cell receptors is associated with poor outcomes for HNSCC patients [46,47].

Myeloid-derived suppressor cells (MDSCs) are another class of inhibitory immune cells present in the TME of almost all solid tumors. MDSCs are a heterogeneous population of immature immune cells comprising early myeloid progenitors, immature dendritic cells (DCs), neutrophils, and monocytes, which negatively regulate the activity of NK cells and induce Tregs [48]. Though difficult to identify due to their diversity, MDSCs were initially identified from HNSCC patients as immature CD34^+^ cells [49,50]. MDSCs inhibit the production of innate inflammatory cytokines such as IL-23, IL-12, and IL-1 by DCs, thereby suppressing antitumor IFN-γ secreting CD4^+^ and CD8^+^ cytotoxic T cells [51]. They regulate T cell activation, migration, proliferation and induce apoptosis by overexpressing immunomodulatory cytokines like IL-10, TGF-β, CD86, PD-L1, TGF-β and suppressing IFN-γ production [52]. MDSCs indirectly suppress the T- cell activation by inducing Tregs, TAMs, and modulating NK cell activity. They also promote angiogenesis and metastasis by producing βFGF, TGF-β and, VEGFA and degrading ECM [53]. Therefore, targeting the inhibitory functions of MDSCs represents a novel avenue for therapeutic intervention in HNSCC tumors.

Regulatory T-cells are immunosuppressive cells with a crucial function in maintaining self-tolerance immune homeostasis (reviewed in [51]). They are also known to regulate CD4^+^ and CD8^+^ T cells, macrophages, B cells, NK cells, and DCs. Based on origin, localization, and marker expression, Tregs are mainly divided into CD25^+^ CD4^+^ Tregs (natural regulatory T cells) that mature in the thymus and peripheral CD25^+^ CD4^+^ Tregs (induced or adaptive Tregs). [54]. These Tregs are known to function by releasing IL-35, IL-10, and TGF-β, inhibiting DC maturation, cytolysis and granzyme/perforin dependent killing of cells, metabolic disruption of effector T cells, and modulation of DC maturation [55]. The genomic and epigenomic differences between HPV**^+ve^** and HPV**^−ve^** HNSCC tumors favor less infiltration of PD-1 and TIM3 co-expressing CD8**^+^** T-cells in HPV**^−ve^** HNSCC [56]. On the contrary, HPV^+ve^ HNSCC tumors are infiltrated with increased Tregs, Tregs/CD8^+^, and CD56^low^ NK cells, CD56^+^ CD3^+^ NKT cells, CD3^+^ T cell, and activated T cells with increased CTLA4 and PD-1 expression and PD-1/TIM3 co-expressing CD8^+^ T cells, suggesting compromised immune system [57]. All these studies suggest heterogeneity in cellular phenotype, function, and location among HPV**^+ve^** and HPV**^−ve^** HNSCC tumors and may potentially be responsible for the varied therapeutic responses.

Besides Tregs, MDSCs, NK, macrophages, neutrophils, platelets, mast cells, adipose cells, and neuroendocrine cells constitute an integral part of the HNSCC TME. In addition to their thrombosis and wound healing activities, thrombocytes or platelets play an important role in tumor biology and inflammation. Besides the secretion of specific granules, viz. dense granules, lysosomes, and α-granules involved in platelet aggregation, platelets also secrete various growth factors in the TME [58]. Interestingly, these granules also contain membranous protein CD63 and lysosomal associated membrane protein 1 & 2 (LAMP1/2), integrin α2β3, p-selectin and glycoprotein-Iβ (GP-Iβ), and secrete molecules like ATP, ADP, Ca^2+^, serotonin, phosphatase into the TME [59]. It is interesting to mention that CD63 and LAMP1/2 membrane proteins help create an acidic environment for acid hydrolases’ optimum activity to degrade ECM [60]. Besides, α-granules also contain many growth factors, a wide variety of chemokines, MMPs, proteins like thrombospondin, fibrinogen, fibronectin, vitronectin, Von Willebrand factor (VWF), and inflammatory proteins that stimulate tumor growth and angiogenesis [61].

Mast cells are another critical component of the immune system regulating both innate and acquired immune response. When mast cells undergo cross-linkage with IgE receptor (FcERI) on their surface, mast cells exocytose many inflammatory mediators including histamine, heparin, prostaglandin D2 (PGD2), leukotriene C4 (LTC4), chondroitin sulfate E, chymase, tryptase, Cathepsin G, carboxypeptidase-A (CPA1), GM-CSF and interleukins into the TME [62]. These cells also secrete fibroblast growth factor-2 (FGF-2), VEGF, MMPs, protease, cytokines, chemokines, and promote proliferation, invasion, and migration of neoplastic cells and angiogenesis [63,64]. Mast cells produce numerous pro-angiogenic factors specific to HNSCC TME, such as FGFβ, TGFβ, tryptase, heparin, and MMPs, to support growth and development [65].In the HNSCC, increased mast cell numbers have been associated with angiogenesis and tumor progression [66].

Neuroendocrine cells release norepinephrine (NE) and epinephrine (E) neurotransmitters. They may either show strong antitumor properties or pro-tumorigenic effects by regulating tumor cell invasion and migration and modulating the immune response. Neurotransmitter substance P (SP) (a member of the tachykinin neuropeptide family), secreted by both tumor and stromal cells, is known to induce many cytokines (IL-1, IL-6, TNF-α). Neurotransmitter SP stimulates tumor cell migration and blocks the integrin β1 mediated adhesion of T cells [67]. In addition, SP also acts as a mitogen factor via a neurokinin-1 receptor (NK-1R), activates protein kinases (PK1 and 2), and promotes cell migration [68], proliferation and protection from apoptosis [69]. Interestingly, both SP and NK-1R are overexpressed and associated with the development and progression of HNSCC [70,71]. Secretion of NE neurotransmitters also inhibits TNF-α synthesis and thereby prevents the generation of CTLs [72]. The α- and β-adrenoreceptors (ARs) for NE and E are overexpressed in the HNSCC cell lines [73], and administration of NE has been shown to increase the proliferation of these cells [74]. A recent study has shown that increased expression of β2-AR promotes EMT in HNSCC cells by activating the IL-6/STAT3/Snail1 signaling pathway [75]. In addition, increased expression of β2-AR was associated with differentiation, lymph node metastasis, and reduced OS of HNSCC patients [75].

Dendritic cells are the most potent antigen-presenting cells (APCs). Through their interaction with lymphoid and myeloid cells, DCs play a vital role in regulating adaptive and innate immune responses during normal and pathophysiological conditions [76]. DCs become immunogenic upon maturation by up-regulation of MHC class II, co-stimulatory molecules, and by secretion of pro-inflammatory cytokines like IL-12, TNF-α, IL-1, and IL-6 [77]. Interestingly, tumor-associated or tumor-treated DCs show low levels of co-stimulatory molecules [78], slow production of IL-12, inhibited antigen-processing machinery (APM), suppressed endocytic activity, and abnormal motility, etc. [79,80]. While higher tumor infiltration of immature DCs is usually observed, increased immature DCs in patients’ peripheral blood with HNSCC, esophageal, lung, and breast cancer have also been reported [81]. Through abortive proliferation, anergy of CD4^+^ and CD8^+^ T lymphocytes or Tregs produce IL-10 and TGF-β and prevent immune response, immature DCs also induce tolerance, thus inhibiting co-stimulatory signals [82,83]. 

Endothelial cells (ECs) play an important role in the development and progression of many tumors [84]. By secreting large amounts of VEGF [85], ECs, in an autocrine manner, induce Bcl-2 expression in the TME micro-vessels and promote angiogenesis and tumor growth [86,87,88]. By regulating the secretion of various CXC chemokines in the HNSCC TME, Bcl-2 is known to enhance invasiveness and the development of recurrent tumors [89,90]. VEGF via IKK/IκB/NF-κB signaling pathway also modulates the expression levels of growth-related oncogene GRO-α (or CXCL1) and interleukin 8 (CXCL8) expression in HNSCC [86] and promotes the development of aggressive tumors. Another study reported that Jagged1, a notch ligand, induced by the growth factors via the activation of mitogen-activated protein kinase-activator protein-1 (MAPK) in HNSCC cells, triggered Notch signaling in adjacent endothelial cells, thus enhancing neovascularization and tumor growth in vivo [91]. Like the ECs, pericytes are an important cellular component of TME and critically important for tumor initiation, progression, and angiogenesis [92]. Pericytes and ECs communicate with each other by paracrine signaling or by chemo-mechanical signaling pathways [93]. By providing mechanical and physiological support to EC, pericytes stabilize vascular walls, promote vessel remodeling, maturation [94,95], regulation of blood flow, and vessel permeability [96]. Although, there are limited studies on the role of pericytes in HNSCC, some studies have shown the presence of abnormal vessels in the OSCC tumor tissue and a reduction of pericyte population in the peritumoral area, thus showing that the pericyte population is significantly affected during OSCC development [97,98,99].

### Extracellular Matrix and Chemokine/Cytokine Activation

The ECM is a 3D network of interwoven macromolecules including glycoproteins, structural fibrous proteins (collagen, elastin, fibronectin, laminin, and tenascin), immersed with enzymes, growth factors, non-cellular components, physical and chemical parameters such as pH, oxygen tension, interstitial pressure, and fluid flux. The ECM provides biophysical, structural, mechanical, and biochemical support to the surrounding cells and helps in-cell adhesion, cell–cell communication, and differentiation [100,101]. Collagen, which constitutes about 30–40% of the total mass of ECM, plays a vital role in cell behavior regulation and development by providing mechanical and structural support and helps in cell adhesion, differentiation, migration, wound repair, and tissue scaffoldings [102,103]. Overexpression of type IV collagen is often observed in HNSCC [104], and collagen XVII, Col15 interaction with integrins has been shown to chemotactically attract HNSCC cells [105]. Notably, Type I collagen has been shown to stimulate the expression of IL-1α, IL-1β, IL-6, TNF-α, and TGF-β in HNSCC [106]. Glycoprotein fibronectin (Fn) produced by fibroblasts and endothelial cells interacts with fibrin, integrins, heparin, collagen, gelatin, and syndecan and promotes tumor progression, migration, invasion, and therapeutic resistance [107]. Peptide hydrolases and MMPs produced by tumor and stromal cells cleave the basement membrane, cell surface receptors, and adhesion molecules and result in the disorganization and deregulation of ECM necessary for invasion and metastasis [108,109]. As in many tumors, ECM proteins such as collagen, laminin, and fibronectin have been shown to promote HNSCC tumor growth, progression, and metastasis [110,111]. Besides, increased expression of fibronectin, tenascin, and decreased expression of laminin, collagen type IV and vitronectin have also been reported to be associated with aggressive HNSCC phenotypes [37,112,113,114]. While the interaction of integrins, particularly α5β1 integrin with fibronectin, and αvβ5 with vitronectin were shown to modulate HNSCC cell behavior, αvβ3-osteopontin, αvβ3-fibronectin, and α5β1-fibronectin interactions are involved in angiogenesis [115]. Overall, ECM plays a very pivotal role in the development and metastasis of tumors by altering the phenotype of stromal or tumor cells, availability of secreted cytokines/chemokines and growth factors, providing acidic and hypoxic conditions for the tumor cells to survive and prevent neoplastic cells from immune attack [116].

## 3. Deregulated Chemokine and Cytokine Expression in HNSCC

Deregulation of cytokines and chemokines is a hallmark of many cancers [18,19]. Using bioinformatics analysis of the Cancer Genome Atlas (TCGA) data, we also observed many cytokines and chemokines deregulated in HNSCC (Figure 2). Consistent with these observations, previous studies have also reported a decrease in Th1 and increase in Th2 cytokine levels [117,118] such as IL-4, IL-6, IL-8, IL-10, GM-CSF, VEGF, prostaglandin E2 (PGE2), and bFGF during the development and progression of HNSCC [119,120,121]. While the increased IL-10, IL-17A, and IL-22 levels, and decreased IFN-γ expressions are collaborated with the loco-regional metastasis [122,123], increased VEGF, FGF, and IL-8 expression contribute to tumorigenesis, metastasis, and HNSCC angiogenesis [16,124,125]. In addition, stromal IL-33 has been shown to promote the enrichment of Foxp3^+^ Tregs and correlated with poor HNSCC prognosis [126]. These studies further showed that stimulation by IL-33 increased infiltration of ST2-expressing Foxp3^+^ GATA3^+^ Tregs (ST2 is the only receptor of IL-33) with increased expression of immune suppressive IL-10 and TGF-β1 [126]. Importantly, IL-1β is known to promote drug resistance by modulating Snail expression, thereby regulating COX-2-dependent E-cadherin expression in HNSCC [127]. Furthermore, TGF-β is known to increase invasion and metastasis by increasing STAT3 expression and malat1/miR-30a interaction in HNSCC [128]. While the increased expression of CCL2 by CAFs enhanced proliferation, invasion and metastasis, and HNSCC tumor growth, the use of specific CCL2 inhibitors significantly reduced tumor burden in vivo [129]. Similarly, increased expression of CCL3 and CCR1 was observed in HNSCC and associated with increased lymph node metastasis [130]. Increased expression of CCL5 in OSCC was also shown to induce MMP-9 secretion and increased cell migration, but the use of siRNA against MMP-9 inhibited CCL5 induced cell motility [131]. Using in vitro and in vivo models, CCL7 has been shown to modulate cytoskeleton re-organization in OSCC, an important regulator of invasion and migration [132]. Use of CCL7 neutralizing antibodies or CCR1 and CCR3 antibodies inhibited invasiveness of OSCC cells [132]. Increased CCL20 or MIP-3α is also associated with increased metastasis and the use of CCL20 siRNA reduced invasive and migratory potential of OSCC cells [133]. CCL21 is a potent stimulator for SCC migration [134], and CCR7 (CCL21 receptor) positive cells have increased capacity to adhere to lymph nodes [134]. In collaboration with these studies, upregulated CCR7 expression in HNSCC has been shown to induce cytoskeletal reorganization, and increasing MMP-9 thereby stimulated migration, invasion, and adhesion [134,135,136,137]. Notably, increased CCR7 expression was correlated with tumor size, clinical stage, recurrence, lymph node metastasis, poor OS, and DFS of HNSCC [132]. Likewise, overexpression and hyperactivity of CCL19/CCL21/CCR7 signaling pathway were positively correlated with lymph node metastasis, and poor prognosis of HNSCC patients [134,138,139].

Like many cytokines, increased CXCL1 expression activates epidermal growth factor receptor (EGFR) signaling and increased human dysplastic oral proliferation [140]. Likewise, CXCL8/CXCR1 and CXCL8/CXCR2 axis are known to induce tumor growth, angiogenesis, motility, and EMT [141]. Consistent with these observations increased expression of CXCL8 and CXCR2 in HNSCC has been shown to promote invasion and migration, and this effect was reversed upon the use of siRNA or blocking antibodies against CXCL8 or CXCR2, respectively [142]. Importantly, CXCL8 polymorphisms are also associated with an increased risk of HNSCC development [143]. While CXCL9 upregulation was reported in the serum of patients with HNSCC compared to healthy controls and associated with poor clinical outcome [144], its downregulation by siRNA resulted in a significant reduction in cell proliferation, migration, and invasion of HNSCC in vitro [144]. Similarly, CXCL12/SDF-1, an α-chemokine via G-protein-coupled CXCR4, regulates stem/progenitor cell trafficking [145]. Of the entire chemokines, CXCL11/CXCL12/CXCR4/CXCR7 axis is the most studied chemokine system in HNSCC [146]. While the expression of CXCL12 (SDF-1α) and CXCR4 progressively increased from oral leukoplakia (OLK) to dysplasia to frank malignancy [147], hyperactivation of this axis in HNSCC is associated with aggressive tumors, regional and distant metastasis, and lower DFS [148]. Furthermore, CXCL12 polymorphism is associated with an increased risk of HNSCC development [149]. Similarly, XCR1/XCL1 axis is known to enhance MMP-2, MMP-7, and MMP-9 secretion and increase proliferation, invasion, and migration of HNSCC cells [150]. All these studies confer the involvement of cytokines and chemokines in the development of aggressive tumors and therefore as important avenues for novel therapeutic intervention in HNSCC (Table 1).

## 4. Chemokine and Cytokine Mediated Signaling Pathways in HNSCC

Chemokines and cytokines exert their effects by activating diverse signaling pathways in HNSCC (Figure 3). Chemokines like TNFα, IL-1, HGF, IFN-α, and their receptors activate MAPK, nuclear factor-kappa-β (NF-kB), and phosphatidylinositol-3 kinase (PI3K)/Akt, signal transducer and activator of transcription (STAT) pathways involved in cell proliferation, survival, invasion, metastasis, and tumor growth [151,182]. NF-kB regulates many genes involved in inflammation and tumor progression. [183,184]. Proinflammatory cytokines like IL-1 and TNF-α by activating IkB kinases (IKKs) and casein kinase 2 (CK2) promote phosphorylation and degradation of NF-kB inhibitors (IkBs). CCL22 is overexpressed in HNSCC and is involved in cell proliferation, migration, invasion, cell transformation, and Tregs infiltration [154]. Secretion of IL-1β by CAFs activates NF-κB signaling in the tumor cells, thereby increasing CCL22 expression [154]. By activating the ERK signaling pathway and phosphorylation of c-Jun/Fos, c-Myc, and E-26-like protein 1 [155], IL-1β also promotes cell survival and tumor progression [185]. Fascin is an actin cross-linking protein that promotes tumor cell invasion [156]. IL-1β activates ERK1/2, JNK, NF-κB, and CREB signaling pathways, increasing Fascin expression and promoting the invasion of HNSCC cells [156]. In addition to IL-1β, increased expression of IL-1α was also correlated with distant metastasis of HNSCC [186]. 

As mentioned earlier, CXCR7 plays an important role in the progression of many cancers. In the HNSCC, CXCR7 activates Smad2/3 signaling, increases TGF-β1 secretion, and results in EMT of HNSCC cells associated with increased invasion and metastasis [187]. TGF-β, TGF-β receptor II and TGF-β-activated kinase 1 (TAK1) are upregulated in HNSCC and result in constitutive hyper-activation of NF-κB with increased cell proliferation, migration, and invasion [188]. TGF-β1 via the TβRII/Smad3 signaling pathway also induces VEGF secretion from HNSCC associated macrophages and helps promote angiogenesis and metastasis [189]. In addition, an alternative TGF-β1-Smad3-Jagged1-Notch1-Slug signaling pathway has been shown to favor tongue squamous cell carcinoma (TSCC) [190]. Similarly, IL-17A was found to promote TSCC by downregulating the expression of miR-23b via the activation of the NF-κB signaling pathway [191]. In addition to TGF-β1, TGF-β2 has been shown to specifically activate MAPK/p38α/β signaling in the bone marrow that causes induction of DEC2/SHARP1 and p27, and the downregulation of cyclin-dependent kinase 4 (CDK4) and results in the dormancy of malignant disseminated tumor cells (DTCs) [192]. IL-6 is known to activate many signaling cascades including JAK/STAT, PI3K/AKT, RAS/MAPK, and Wnt signaling pathways affecting angiogenesis and metastasis [193,194,195,196], and to induce a dysfunctional immune response. Constitutive activation of IL-6/STAT3 signaling is associated with reduced (OS) in p16^−ve^ HNSCC [197]. It was recently shown that the IL-6 induced STAT3 signaling pathway promoted immunosuppressive HNSCC TME by upregulating PD-1/PD-L1 expression [198]. However, inhibition of STAT3 signaling pathway downregulated PD-1/PD-L1 expression and improved immune surveillance in TGFβr1/PTEN 2cKO mouse model of HNSCC [198]. IL-6 was also shown to promote HNSCC tumorigenesis by activating fibroblasts and increasing tumor cells–CAF crosstalk [199]. Like the IL-6/STAT3 signaling axis, activation of the STAT1 signaling pathway by IFN-α promotes immunosuppression in HNSCC [200,201]. The underlying molecular mechanisms revealed increased PD-L1 and RIG-I expression in tumor cells [200,201], and PD-1 in immune cells [200] via IFNαR1 activation. IL-8 is upregulated in HNSCC and affects pathways involved in inflammation. The RAS/MAPK signaling pathway is critical for carcinogenesis and regulates inflammation, cell proliferation, survival, and tumor growth [202]. Using in vitro studies, IL-8 was shown to regulate inflammatory response by activating both NF-κB and MAPK signaling pathways in HNSCC [203]. In addition, IL-8 also activates the CXCR1/2-mediated NOD1/RIP2 signaling pathway, thereby facilitating the formation and progression of HNSCC [161]. SDF-1/CXCL12-CXCR4 axis has been shown to activate the Akt/PKB and ERK1/2 signaling pathway and induce directional tumor cell migration, supporting its role in invasion and metastasis of HNSCC cells [204]. MCP-1/CCL2, is a potent monocyte-attracting chemokine which helps recruit monocyte to the tumors [205], modulate pro-survival signals and promote HNSCC progression [206].

## 5. Chemokines and Cytokines Promote Aggressive HNSCC Phenotype

Cytokines, chemokines, and their receptors are important players of TME and are known for initiation, promotion, progression, metastasis, and development of aggressiveness [207]. For example, upregulation of CXCR7 increased cell migration and invasion through the Smad2/Akt signaling pathway, promoted lymph node metastasis [208], and is associated with the aggressive phenotype of HNSCC [187]. Further studies showed that CCR7 in HNSCC is upregulated by NF-kB and AP1 and contributes to metastatic phenotype [209]. Infiltrated macrophages are well-known contributors to aggressive HNSCC [210]. The underlying mechanism revealed the involvement of the CCL2/EGF positive feedback loop. The tumor cell-derived CCL2 transforms monocytes into M2-like macrophages, resulting in the increased production of EGF, which activates EGFR signaling in the tumor cells promoting the formation of invadopodia associated with increased HNSCC cell motility [210]. Like CCL2, the role of the CXCL12 (SDF-1α)/CXCR4 axis in the metastatic processes of HNSCC has been explored in many studies. Activation of the CXCL12/CXCR4 axis enhances cell adhesion and MMP-9 secretion, thereby increasing HNSCC metastasis [211]. The CXCL12/CXCR4 axis, by upregulating MMP-13 via activation of ERK_1/2_/AP-1 signaling pathway, also increased invasion and metastasis of laryngeal and hypopharyngeal SCC (LHSCC) [147,212]. In support of these studies, CXCR4 upregulation in HNSCC is confined to tumor nests, but not in the stroma [213,214]. CXCL5 and CXCL9 chemokines are essential determinants of tumor development and malignancy. Like SDF-1α, overexpression of CXCL5 and CXCL9 induced invasion and migration of HNSCC cells and aggravated HNSCC phenotype [144,215,216]. In a paracrine manner, CAF secreted CCL11, and IL-33 promoted migration, invasion, and aggressive HNSCC phenotype of cells [217,218]. Similarly, CCL21/CCR7 axis has been shown to promote MMP-9 release, stimulate tumor cell survival, adhesion, invasion, and metastasis in HNSCC [134,135,136,137]. IL-8 and its receptors CXCR1 and CXCR2 are overexpressed in HNSCC and involved in progression, metastasis, and aggressive tumor phenotype [161,219]. The underlying mechanisms revealed inactivation of PTEN and activation of the STAT3 signaling pathway by IL-8/CXCR1/2 axis in promoting aggressive HNSCC phenotype [219]. In addition to IL-8, migration inhibitory factor (MIF) from tumor cells induces CXCR2-dependent chemotaxis, improved neutrophil survival, and release of CCL4 and MMP-9, helping develop aggressive HNSCC phenotype [220]. Besides IL-8, the STAT3 signaling pathway is also activated by the IL-6 cytokine known to be upregulated in most HNSCC patients. The IL-6 mediated STAT3 activation upregulates many downstream target genes involved in proliferation, invasion, migration, and EMT of HNSCC [193], suggesting its role in aggressive tumor behavior. IL-6 promoter has aryl hydrocarbon receptor (AhR), suggesting the involvement of AhR in translational regulation of IL-6 and development of aggressive HNSCC. The use of AhR antagonists has been shown to reduce IL-6 expression and decreased the aggressive phenotype of HNSCC cells [221]. Moreover, defects in TGF-β signaling are found to be associated with the growth and development of HNSCC [222]. A recent study has reported overexpression of TGF-βRII in HNSCC and found this to be inversely correlated with local disease aggressiveness [223].

## 6. Conclusions and Future Directions

Despite significant advances in the treatment modalities, the prognosis of HNSCC patients has not changed considerably in decades, the underlying reasons being the aggressive tumor behavior associated with local and distant metastasis at the time of diagnosis, and intrinsic and acquired resistance to the currently available therapies. The HNSCC TME is a very complex structure with interplay and convergence of several signaling pathways. Cytokines and chemokines, as important entities of these interplays, contribute to tumor growth, aggressiveness, and metastasis in HNSCC. Therefore, exploring the potential of chemokines and cytokines can aid in developing novel therapeutic approaches for the treatment of HNSCC. The use of chemokine receptor antagonists or inhibitors and anti-chemokine antibodies can also serve as an adjuvant chemotherapy alternative in HNSCC. As mentioned above, the CXCL11/CXCL12/CXCR4/CXCR7 axis is the most studied chemokines system in HNSCC, and it can serve as an important therapeutic target. In addition, CCL19/CCR7, CCL5/CCR5, and CCL2/CCR2 axis are important targets for therapeutic intervention and would help improve HNSCC outcomes. Studies highlighting the contribution of individual pathways and their predominance in response to a particular mutation will help in achieving an optimum therapeutic outcome in HNSCC.

Moreover, IFN-γ and IL-7 can be used as risk markers in neck metastasis due to their downregulation, specifically in HNSCC cases with nodal metastasis [175]. As IFN has been found to exhibit anti-proliferative properties, it can be used as an immunomodulatory agent. Another immunotherapy approach for the treatment of HNSCC is the application of recombinant cytokines which can induce a targeted manipulation of the immune system. Stimulation of immune cells that can alter the cytokine profile in HNSCC and diminish its immunosuppressive effects is another interesting approach. Typically, delay in diagnosing HNSCC requires surgical treatment with the combination of radio- or chemotherapy. In this context, salivary cytokine can serve as diagnostic biomarker and predict CRT outcome in HNSCC [224]. Furthermore, elevated levels of IL-6, IL-8, VEGF, HGF, and GRO-1 found in HNSCC patients with poor survival indicate that targeting this pathway could be of therapeutic significance [225]. Moreover, IL-15 can act as a therapeutic target in HNSCC as is reported to be a powerful stimulator of NK and CD8^+^ T cell function. Recently, a phase I clinical trial was conducted to determine the maximum tolerable dose of recombinant human IL-15 (rhIL15) in patients with advanced solid tumors including HNSCC [124]. Although the clinical trial showed no objective clinical responses, few patients were found to show disease stabilization after IL-15 administration [124]. Another strategy for restoring antitumor immune functions in HNSCC is the use of a primary-cell-derived biologic known as IRX-2. IRX-2 is a natural cytokine biologic derived from peripheral blood mononuclear cells (PBMCs) that contain active components such as IL-2, IL-1β, TNF-α, and IFN-γ [226]. IRX-2 helps overcome tumor-mediated immunosuppression by acting on multiple immune cells such as DCs, NK, and T cells. Several promising phase I and phase II clinical trials have been conducted on IRX-2, thus suggesting that it can be used as an immunotherapy target for the treatment of HNSCC as well as other malignancies [226]. Therefore, in the future, cytokine-based immunotherapies must focus on combining strategies/schemes that enhance antitumor responses and suppress protumorigenic immune cells. Moreover, new approaches such as vector delivery or modified recombinant proteins can improve cytokine targeting and enhance the efficacy of cytokine-based immunotherapies in cancer [227]. In conclusion, chemokines and cytokines are the essential players for HNSCC pathogenesis and targeting their complex networks could become therapeutic strategies explicitly targeting HNSCC.

## Figures and Tables

**Figure 1 ijms-22-04584-f001:**
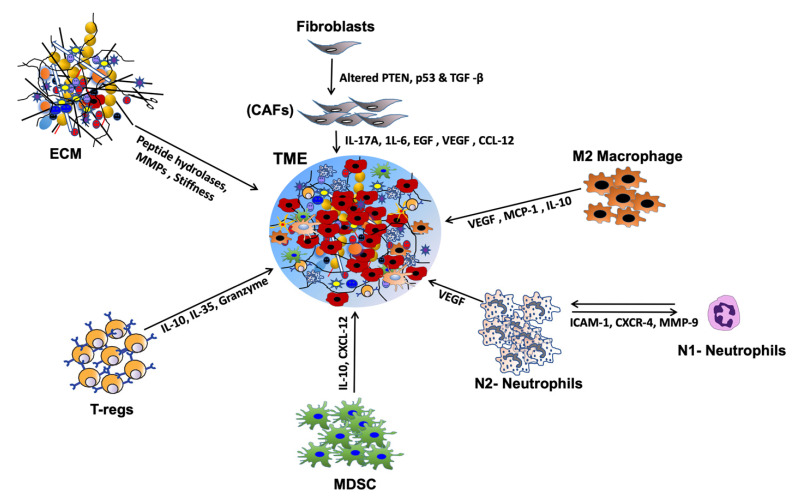
Chemokine and cytokine-mediated crosstalk in head and neck squamous cell carcinoma (HNSCC) tumor micro-environment (TME). Cytokines and chemokines secreted by a variety of stromal cells affect tumor cell growth, proliferation & metastasis in many ways. By inducing immune-suppressive TME, they promote immune evasion and metastasis. Many chemokines and cytokines help degrade extracellular matrix (ECM) proteins, induce angiogenesis, and thereby promote invasion and metastasis.

**Figure 2 ijms-22-04584-f002:**
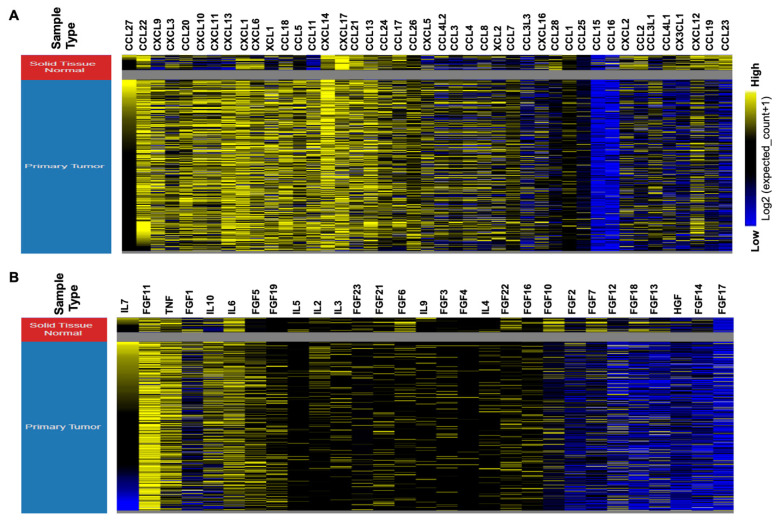
Deregulated chemokines and cytokines in HNSCC. The heat maps showing deregulated expression of (**A**) chemokines and, (**B**) cytokines in HNSCC patients. The heat maps were constructed through data mining in the HNSCC TCGA database by using the UCSC Xena browser (http://xena.ucsc.edu (accessed on 9 February 2021)) (adjacent normal, *n* = 44, tumor tissue, *n* = 518 and metastatic = 02) samples).

**Figure 3 ijms-22-04584-f003:**
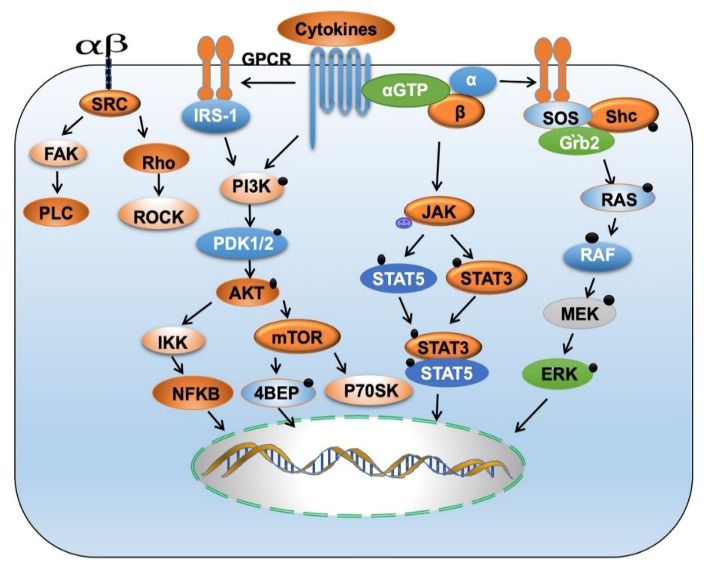
Cytokines/Chemokines activate signaling pathways. Chemokines/cytokines via transmembrane protein GPCR activate an array of signaling pathways like PI3K/AKT/mTOR, JAK/STAT, RAS/RAF, integrin mediates SRC/FAK and Rho/RAC. Deregulation of these signaling pathways is known to promote initiation, progression, and metastasis in HNSCC.

**Table 1 ijms-22-04584-t001:** Deregulated Chemokines/Cytokines and their Therapeutic Targeting in HNSCC.

S.No	Chemokine/Cytokine	Expression	Signaling Pathway(s) Activated	Interacts with Cells	Involved in	Targeted OR Can Be Targeted by	References
1	IL-1	Up	MAPK/ERK_1/2_, NF-KB, PI3K/AKT, & JAK/STAT	Tumor, T-cells, TAMs & macrophages	Recruitment of TAMs, MDSCs and Tregs	Recombinant IL-1R antagonist (anakinra)	[151,152]
2	IL-1β	Up	NF-kB, ERK_1/2_, JNK, CREB	Endothelial & leukocytes	Increase integrin expression	Lenti virus mediated shRNA	[153,154,155,156,157]
3	IL-4	Up	MAPK	Tumor, TAMs & endothelial cells	Promote tumor growth, angiogenesis & immuno-suppression	rIL-4 with Pseudomonas exotoxin (PE) targeting IL-4R	[158]
4	IL-6	Up	JAK/STAT, PI3K/AKT, RAS/RAF/MEK/ERK_1/2_ & Wnt	Tumor, Th17 cells & CAFs	Support tumor growth, immune evasion & CAF activation	Humanized antiIL-6R antibody (Tocilizumab)	[159,160]
5	IL-8	Up	NF-kB, MAPK, CXCR1/2/NOD1/RIP2	Tumor, endothelial cells & neutrophils	Promotes tumor invasion, angiogenesis & recruit neutrophils to the TME	Humanized anti- IL-8 antibody (HuMax-IL8)	[161,162]
6	IL-10	Up	JAK/STAT3	T cells, TAMs	Suppress T-cell proliferation and promote immuno-suppression	-	[163,164]
7	IL-15	Up	JAK/STAT3, PI3K/AKT	CD8+ T cell & NK cell	Stimulate NK and CD8+ T cell function	Recombinant human IL-15 (rhIL15)	[124]
8	IL-33	Up	RAS/RAF//MEK/ERK_1/2_ and JNK	Tregs	Enrich FOXP3^+^ Tregs	Anti-IL-33 antibodies	[125,129,165]
9	CCL-2	Up	MAPK, PI3K/AKT	Monocytes, TAMs	Recruits TAMs and monocytes	CCL-2 inhibitor mNOX-E36 or neutralizing antibody CNTO88	[166]
10	CCL-7	Up	NF-kB and MAPK	DCs, NK and T cells	Recruitment of TAMs and CAFs proliferation	CCL-7 neutralizing antibodies	[132,167]
11	CCL-20	Up	NF-kB, RAS/RAF/MEK/ERK_1/2_	NK, TAMs and Tregs	Recruit TAMs & Tregs	Anti-CCL20 (WO2017011559A1) or CCR6 antibodies	[133,168]
12	CXCL-1	Up	PI3K/AKT/mTOR, RAS/RAF/MEK/ERK, and NF-ĸB	MDSCs, TAMs, CAFs	Recruit MDSCs to TME and promote metastasis	Small molecule inhibitor (C29) against CXCR1	[140,169]
13	CXCL-8	Up	PI3K/AKT, FAK/Src, Rho/GTPase and MAPK	CSC, endothelial cells	Promote angiogenesis, and CSC proliferation	Antibodies (ABX-CXCL8, HuMax CXCL8) or CXCR1/2 inhibitor Reparixin	[142,170]
14	CCR-1	Up	RAS/RAF/MEK/ERK, AKT-mTOR, JAK/STAT3	MDSCs, TAMs, T cells	Tumor infiltration of MDSCs and Treg cells	CCR-1 inhibitor MLN3897	[169,170,171]
15	CXCR-2	Up	MAPK/MEK/ERK_1/2_, NF-kB, PI3K/AKT and JAK/STAT3	Cancer cells, TAMs, Monocytes, T cells	Increase cell proliferation	Antibodies (ABX-CXCL8, HuMax CXCL8) or CXCR-2 inhibitor Reparixin	[162,172]
16	GM-CSF	Up	JAK/STAT, SRC kinase, MAPK/MEK/ERK_1/2_, and PI3K/AKT	Tumor cells, DC	DC differentiation, TAM & Treg function and favour IL9-producing Th (Th9) cells	Recombinant GM-CSF or GM-CSF-based DNA vaccines	[18,19,173]
17	βFGF	Up	FGFR, JAK/STAT, RAS/RAF/MEK/ERK	Tumor cells, CAFs and endothelial cells	Promote cell proliferation and tumor growth	FGFR inhibitors or antibodies (MFGR1877S, BAY 1179470)	[174]
18	IFN-γ	Down	IFN-γR_1/2_/JAK-STAT, PI3K/AKT/mTOR, IFN-γ/ICAM1-PI3K-Akt-Notch1	Tumor cells, DC, T cells, Tregs, Macrophages,	Recruits NK cells to the TME & modulates their activity	rIFN-γ	[122,123,175,176]
19	TGF-β	Up	TGFβRII/SMAD3, TGFβ/TAK1, NF-κB	Tumor cells, Macrophages, Tregs, NK cells, MDSCs	DC dysfunction, TAMs formation, suppression of NK cells, MDSC recruitment	Anti TGF-β antibody	[18,19,128,177,178]
20	TNF-α	Up	MAPK, PI3K/AKT, NF-kB, JAK/STAT, FAK/Src	Tumor cells, CAFs	Increase angiogenesis, invasion, and metastasis	-	[18,19,179,180]
21	VEGF	Up	PI3K/AKT/mTOR, RAS/RAF/MEK/ERK	Endothelial cells	Controls vascular permeability & angiogenesis	Anti-VEGF (Bevacizumab) or anti-VEGFR (Ramucirumab) antibodies	[181]

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
