# Peer review of "Chemokine-Cytokine Networks in the Head and Neck Tumor Microenvironment"

_ijms, 2021, doi:10.3390/ijms22094584_

Round 1

Reviewer 1 Report

This manuscript is a new addition to the long list of recent reviews focusing on the effects of Chemokine/Cytokine in the tumor microenvironment (TME) and their abilities to recruit and activate immune cells to promote tumor development, progression and metastasis.  The objective is to summarize the current understanding of the complex cytokine/chemokine networks in the head and neck squamous cell carcinomas, their role in activating diverse signaling pathways to promote tumor growth as well as therapeutic resistance.  The manuscript is well-written and the recent findings are well analyzed and presented.  A drawback, however, is the lack of specificity as it relates to HNSCC “specific” pathways.  Apart of the introduction, the authors could easily replace head and neck with prostate, breast or pancreatic cancer and the review will sound the same. Instead of a laundry list of the effects of cytokine/chemokine in the recruitment activation of different cell types to the TME, the authors could shorten the manuscript by summarizing recent studies relevant to Head and neck tumorigenesis.    Nonetheless, this manuscript is very important and should be published in a shorter version.

Author Response

Reviewer #1

This manuscript is a new addition to the long list of recent reviews focusing on the effects of Chemokine/Cytokine in the tumor microenvironment (TME) and their abilities to recruit and activate immune cells to promote tumor development, progression and metastasis.  The objective is to summarize the current understanding of the complex cytokine/chemokine networks in the head and neck squamous cell carcinomas, their role in activating diverse signaling pathways to promote tumor growth as well as therapeutic resistance.  The manuscript is well-written and the recent findings are well analyzed and presented.  A drawback, however, is the lack of specificity as it relates to HNSCC “specific” pathways.  Apart of the introduction, the authors could easily replace Head and neck with prostate, breast or pancreatic cancer and the review will sound the same. Instead of a laundry list of the effects of cytokine/chemokine in the recruitment activation of different cell types to the TME, the authors could shorten the manuscript by summarizing recent studies relevant to Head and neck tumorigenesis.    Nonetheless, this manuscript is very important and should be published in a shorter version.

Response: We appreciate the feedback and suggestions from the reviewer. We have removed the unnecessary literature from the revised manuscript and mainly focussed on HNSCC. We also combined the subsections of the HNSCC tumor microenvironment section for better flow and specificity.

Reviewer 2 Report

Nisar et al. wrote a very detailed and comprehensive review about the chemokine-cytokine network in HNSCC.

Since the focus of this review is the role of these molecules in HNSCC a chapter should be given about actual approaches to develop specific drugs targeting chemokines or cytokines for cancer therapy especially for therapy of HNSCC.

Instead of Fig. 2 a table should be presented listing the various chemokines and cytokines with information about e. g. deregulation in HNSCC, role in which pathway, interplay with CAFs and various immune cells and (very important) targetability for (potential) therapeutics.

Author Response

Reviewer #2

Nisar et al. wrote a very detailed and comprehensive review about the chemokine-cytokine network in HNSCC.

Since the focus of this review is the role of these molecules in HNSCC a chapter should be given about actual approaches to develop specific drugs targeting chemokines or cytokines for cancer therapy especially for therapy of HNSCC.

Response: We thank the reviewer for the helpful suggestions and comments. To date, only six drugs have been FDA-approved for the treatment of Head and Neck cancer,  including five conventional CT drugs (cisplatin, methotrexate, 5-fluorouracil [5-FU], bleomycin and docetaxel) and one targeted agent (cetuximab). Poor survival and considerable morbidity of current treatments suggest the need for new therapeutic modalities that can improve outcomes. So far, there is no approved drug targeting cytokines/chemokines or that contains these as a tool to improve antitumor immunity of HNSCC patients. Strategies for relieving immunosuppression and restoring antitumor immune functions could benefit HNSCC patients. In this direction, progress has been made with IRX-2, currently undergoing a Phase IIB INSPIRE trial evaluating the IRX-2 regimen as a stand-alone therapy for activating the immune system to recognize and attack tumors. IRX-2 is a primary cell-derived biologic consisting of physiologic levels of T-helper type 1 cytokines (IL2, IL1β, IFNγ, and TNFα) produced by stimulating peripheral blood mononuclear cells of normal donors with phytohemagglutinin. Besides this, there could be other approaches using cytokines/chemokines as therapeutic agents, and more details are given in the conclusion and future directions section of the revised manuscript.

Instead of Fig. 2 a table should be presented listing the various chemokines and cytokines with information about e. g. deregulation in HNSCC, role in which pathway, interplay with CAFs and various immune cells and (very important) targetability for (potential) therapeutics.

Response: We thank the reviewer for this suggestion. A detailed table (Table 1) has been included in the revised manuscript.